# *GmANKTM21* Positively Regulates Drought Tolerance and Enhanced Stomatal Response through the MAPK Signaling Pathway in Soybean

**DOI:** 10.3390/ijms25136972

**Published:** 2024-06-26

**Authors:** Yue Zhao, Sinan Wang, Xiaofei Ma, Yu He, Jingwen Zhou, Shuang Jiao, Jianing Xun, Xiaoyu Kong, Xiaoxia Wu, Xi Bai

**Affiliations:** Northeast Agricultural University, Harbin 150001, China; zy18714391278@163.com (Y.Z.); 17542166925@163.com (S.W.); mxfdhr@163.com (X.M.); heyu1063825189@163.com (Y.H.); 18846084378@163.com (J.Z.); 18724420808@163.com (S.J.); xjn15833336029@163.com (J.X.); 18800454769@163.com (X.K.)

**Keywords:** *Glycine max*, ankyrin repeat protein, *GmANKTM21*, drought stress

## Abstract

Drought stress is one of the significant abiotic stresses that limit soybean (*Glycine max* [L.] Merr.) growth and production. Ankyrin repeat (ANK) proteins, being highly conserved, occupy a pivotal role in diverse biological processes. *ANK* genes were classified into nine subfamilies according to conserved domains in the soybean genome. However, the function of ANK-TM subfamily proteins (Ankyrin repeat proteins with a transmembrane domain) in the abiotic-stress response to soybean remains poorly understood. In this study, we first demonstrated the subcellular localization of GmANKTM21 in the cell membrane and nucleus. Drought stress-induced mRNA levels of *GmANKTM21*, which encodes proteins belonging to the ANK-TM subfamily, Transgenic *35S:GmANKTM21* soybean improved drought tolerance at the germination and seedling stages, with higher stomatal closure in soybean, lower water loss, lower malondialdehyde (MDA) content, and less reactive oxygen species (ROS) production compared with the wild-type soybean (Dongnong50). RNA-sequencing (RNA-seq) and RT-qPCR analysis of differentially expressed transcripts in overexpression of *GmANKTM21* further identified potential downstream genes, including *GmSPK2*, *GmSPK4*, and *GmCYP707A1*, which showed higher expression in transgenic soybean, than those in wild-type soybean and KEGG enrichment analysis showed that MAPK signaling pathways were mostly enriched in *GmANKTM21* overexpressing soybean plants under drought stress conditions. Therefore, we demonstrate that *GmANKTM21* plays an important role in tolerance to drought stress in soybeans.

## 1. Introduction

Soybeans [*Glycine max* (L.) Merr.] are an essential source of protein, oil, carbohydrates, and micronutrients both in the human diet and in animal feed. However, soybeans are highly vulnerable to abiotic stresses, including drought, high temperatures, hail, and other extreme weather events [1,2,3]. As a fundamental crop on a global scale, soybeans are profoundly vulnerable to water scarcity, leading to considerable reductions in productivity worldwide. Drought conditions, acknowledged as a predominant factor impeding soybean growth and maturation, present a considerable challenge to its cultivation [4]. Therefore, improving drought tolerance in soybeans is necessary.

The ankyrin repeat (ANK) domain, comprised of multiple units each typically containing 33 amino acid residues, plays a crucial role in plant growth and development [5,6]. To analyze the function and domain distribution of *ANK* genes, *ANK* genes were classified into different subfamilies based on their domain composition: ANK-U (unique ANK domain), ANK-TM (transmembrane domain), ANK-PK (containing the protein kinase domain), ANK-ZnF (zinc-finger), ANK-BPA (the BAR, PH and ArfGap domains), ANK-ACBP (Acyl-CoA-binding protein), ANK-GPCR (GPCR-chapero-1), ANK-IQ (calmodulin-binding domain) and ANK-O (containing different domains). Among these genes, the transmembrane domain (ANK-TM) was the second most abundant subfamily [7,8].

*ANK* genes are essential for various biological processes, as evidenced by the *AKR* code containing a protein encoding five anchor protein repeats. Loss of *AKR* function leads to embryonic defects, along with severe damage to chloroplasts and mesophyll cells in homozygous *akr/akr* plant lines [9]. Specifically, the *ANK6* ankyrin repeat protein is indispensable for the fertilization process in *Arabidopsis thaliana*, as its dysfunction results in embryo demise. Heterozygous *+*/*ank6* plants exhibit significantly smaller unfertilized atrophous ovules compared to wild-type ovules, leading to a marked decrease in the fruiting rate [10].

*ANK* family genes also play a crucial role in plant stress responses. Studies have shown that *LR14A*, a representative of the ANK-TM protein family, is involved in disease resistance in wheat. Similarly, the wheat stripe rust resistance gene *YrU1* encodes an NLR protein with an integrated ANK structural domain. The ANK structural domain of YrU1 is derived from the ANK-TM protein, suggesting that ANK-TM proteins are involved in plant disease resistance [11]. Ankyrin repeat (ANK) *C3HC4*-type genes encode XB3-like proteins. Proteins from the *XB3*-like gene family are found in different plant species and are involved in a variety of developmental processes. *XBAT35* belongs to a subgroup of Arabidopsis ankyrin repeat-containing E3 ligases, and *XBAT35* transcript levels are modulated by temperature and high salinity stress [12]. Furthermore, research indicates that the ankyrin repeat sequence *CaKR1*, when overexpressed in tomatoes, can enhance the salt stress tolerance of transgenic tomato plants [13].

Genome-scale analyses of *ANK* genes from some plant species, including Arabidopsis and rice, have shown that *ANK-TM* accounts for 40% or more of the total number of identified *ANK* genes. Despite the large number of genes in the *ANK-TM* subfamily, it is poorly understood [6,14,15], making it particularly important to study the soybean *ANK*-*TM* subfamily genes.

## 2. Results

### 2.1. Subcellular Localization

To determine the subcellular localization of GmANKTM21, a *35S*-controlled GmANKTM21-green fluorescent protein chimera (*CaMV35S::GmANKTM21-GFP*), a *GmANKTM21* promoter-controlled GmANKTM21-green fluorescent protein chimera (*proGmANKTM21::GmANKTM21-GFP*), and a control vector (*CaMV35S::GFP*) were transformed into Agrobacterium tumefaciens strain GV3101 and infiltrated into tobacco, and it was found that CaMV35S::GmANKTM21-GFP and proGmANKTM21::GmANKTM21-GFP fusion proteins were both localized in the cell membrane and nucleus (Figure 1).

### 2.2. The Overexpression of GmANKTM21 Enhanced the Tolerance of Soybeans to Drought during Germination

To confirm the function of *GmANKTM21* in soybean drought tolerance, we generated two T_3_-generation *GmANKTM21*-OE (overexpression) transgenic lines under the control of the *CaMV35S* promoter (Figure 2A). In comparison with DN50 (WT) plants, expression levels of *GmANKTM21* in Line2 and Line13 were significantly higher, by 53-fold in Line2 and 80-fold in Line13 (Figure 2B). The importance of *GmANKTM21* in plant resistance to drought was first evaluated using a germination assay.

Under normal conditions, Line2 and Line13 had longer hypocotyls and higher fresh weights compared to DN50. When exposed to 7.5% PEG6000, the growth of all soybean plants was found to be significantly inhibited, while Line13 grew relatively well. There were no significant differences in growth phenotypes at 15% PEG6000 (Figure 2C–E). Based on the data acquired, it became apparent that under a lower concentration of PEG6000 (7.5%), the germination rates of all three genotypes closely resembled those of their untreated counterparts (0%). However, discernible inhibitory effects of PEG6000 on soybean seed germination were evident at concentrations of 15% and 22.5%. Under these stress-inducing conditions, Line2 and Line13 seeds exhibited significantly heightened germination rates compared to DN50. Line2 and Line13 showed higher germination potential compared to DN50 under normal conditions, and Line13 showed higher germination potential under 7.5% and 15% PEG6000 conditions (Figure 2F,G).

### 2.3. Overexpression of GmANKTM21 Enhanced Drought Tolerance in Soybean Seedlings

In order to further evaluate the drought tolerance of *GmANKTM21*-OE in soybeans, two sets of drought treatments were conducted on Lines2 and Line13. These treatments consisted of exposing the plants to 15% PEG6000 for 24 h, water deficit conditions for three days, and rehydration. Significant phenotypic differences were observed under the drought treatments (Figure 3A,C), with Line2 and Line13 plants showing lower leaf wilting compared to DN50 (Figure 3B,D), suggesting that they are more resistant to drought stress.

### 2.4. Overexpression of the GmANKTM21 Gene Enhanced the Water Retention Ability of Soybean

Since plant water retention capacity contributes to drought resistance [16], we measured the fresh weights of the detached second trifoliate leaves of DN50 and *GmANKTM21*-OE plants over 5 h following their detachment from the plants. Compared to that in DN50, water loss was slower in the leaves from *GmANKTM21*-OE plants (Figure 4A,B). We also observed the stomata on the second trifoliolate leaves of DN50 and *GmANKTM21*-OE plants during drought treatment. Before drought treatment, the stomatal aperture of *GmANKTM21*-OE plants was not significantly different from that of DN50 plants. After drought treatment, the stomatal aperture of *GmANKTM21*-OE plants was significantly smaller than that of DN50, but the stomatal density was not significantly different (Figure 4C–E).

The accumulation of cuticular wax can enhance the water-retaining capacity of plants [17]. We examined the deposition of epidermal wax crystals on the leaf surfaces of DN50 and *GmANKTM21*-OE plants. No significant change in the number of Line2 and Line13 epidermal wax crystals was observed by scanning electron microscopy (SEM) compared to DN50 (Figure 4F). The wax layer directly determines cuticle permeability [18], so the cuticle permeability of *GmANKTM21*-OE and DN50 was examined. The results showed that there was no significant difference in the degree of staining by toluidine blue (TB) in the leaves of *GmANKTM21*-OE compared to DN50 (Figure 4G).

These results suggest that overexpression of *GmANKTM21* enhances drought tolerance by adjusting stomata to enhance water retention in plants, rather than altering leaf cuticle and wax properties.

### 2.5. Overexpression of the GmANKTM21 Gene Enhanced the Antioxidant Capacity of Soybean

To further investigate the role of *GmANKTM21* in regulating drought tolerance, we measured superoxide dismutase (SOD), peroxidase (POD), and catalase (CAT) activities as well as the content of MDA, to reflect the response ability of plants for resilience. The enzymatic activities of SOD, POD, and CAT scavenge intracellular ROS and reduce H_2_O_2_ production, thereby enhancing drought tolerance. The SOD, POD, and CAT activities of DN50 plants were significantly lower than those of the *GmANKTM21*-OE plants (Figure 5A–C). The MDA content is used to indicate lipid peroxidation products and to reflect the extent of plant damage caused by adversity. Compared with the WT plants, the MDA content in *GmANKTM21*-OE plants significantly decreased after drought stress (Figure 5D). Finally, we stained soybean leaves with nitro blue tetrazolium (NBT) and 3,3-diaminobenzidine (DAB) to detect H_2_O_2_ content under normal or drought treatment conditions in *GmANKTM21*-OE lines and DN50 plants. No substantial difference was observed under normal conditions with NBT and DAB staining; however, the color depth of DN50 plants was substantially higher than that of the *GmANKTM21*-OE lines under drought treatment. In contrast, the leaf color depth of the *GmANKTM21*-OE lines was substantially lower than that of DN50 plants (Figure 5E,F). These results suggest that *GmANKTM21* overexpression in soybeans enhanced drought tolerance by reducing the MDA content and increasing the enzyme activities of SOD, POD, and CAT to reduce ROS production.

### 2.6. Studies on Downstream Genes of GmANKTM21 Gene Regulation

To further investigate the drought resistance mechanism of *GmANKTM21*, we performed a comparative transcriptome analysis using leaves from plants of *GmANKTM21*-OE lines and DN50 plants. Based on the RNA-seq data, we identified differential expression genes (DEGs) in the *GmANKTM21*-OE plants compared to DN50. To examine the functions of these DEGs, KEGG pathway analysis was performed. KEGG enrichment analysis showed that the differential genes were mainly enriched in the plant MAPK signaling pathway, starch and sucrose metabolism, and biosynthesis of secondary metabolites (Figure 6A).

The MAPK signaling pathway is involved in plant growth and development, phytohormone signal pathways, and ABA-regulated signal pathways consisting of PYR/PYL/RCAR-PP2C-SnRK2 [19,20]. Based on these results, we suggest that *GmANKTM21* plays an important role in the drought stress response by regulating various drought-related genes and may be involved in the MAPK signaling pathway. To further investigate whether the role of *GmANKTM21* in drought tolerance is associated with the MAPK signaling pathway, we analyzed the expression of genes that function in drought tolerance and genes that are related to the MAPK signaling pathway in DN50 and *GmANKTM21*-OE plants. The results showed that *GmSPK2*, *GmSPK4*, *GmSnRK2.6*, *GmABI5*, and *GmABF4* were up-regulated in Line2 and Line13 after drought treatment. However, in DN50, the expression levels of the *GmSPK2*, *GmSPK4*, and *GmSnRK2.6* genes did not change significantly before and after drought treatment. In addition, *GmCYP707A1* gene expression in Line2 and Line13 plants was significantly lower than that of DN50 before and after drought treatment (Figure 6B–G). In the light of previous studies, it has been observed that the *GmSPK2* and *GmSPK4* genes can be activated in response to hyperosmotic and drought stresses, exerting resistance through pertinent mechanisms [21]. Additionally, the *GmCYP707A1* gene, a member of the *CYP707A1* family, plays a role in the degradation of ABA [22]. Drawing upon the outcomes of our current experiment, we suggest that the *GmANKTM21* gene governs plant drought resistance by modulating the expression of various genes within the MAPK signaling pathway, including *GmSPK2*, *GmSPK4*, and *GmCYP707A1*.

## 3. Discussion

Drought significantly impacts soybean production and quality. ANK proteins facilitate protein-protein interactions, which are crucial for defense responses and, occasionally, growth and development. Understanding ANK protein function sheds light on plant stress responses, aiding environmental adaptation research in plants.

Water is essential for seed germination; however, while other conditions may be ideal, drought stress inhibits the imbibition of seeds and, consequently, hinders germination [23]. Our results revealed improvements in seed germination percentage, root growth, and drought tolerance in the transgenic soybean with *GmANKTM21* in comparison to those of WT plants under drought stress simulated by PEG6000 (Figure 2 and Figure 3), and our results showed that overexpression of *GmANKTM21* in soybeans could also improve plant tolerance to drought stress.

Stomatal closure, recognized as an early response to drought stress, is commonly utilized as a crucial indicator for swiftly assessing plant water status [24,25]. Our investigation validated that the overexpression of *GmANKTM21* facilitates prompt stomatal closure, thereby diminishing transpiration and mitigating leaf water loss during drought conditions (Figure 4C–E). Despite the pivotal role of cuticular waxes in providing the initial defense against environmental stresses [26,27,28], our findings indicate that the overexpression of *GmANKTM21* does not significantly influence the quantity of wax or keratin (Figure 4F,G). This underscores that *GmANKTM21* primarily attenuates water loss by enhancing stomatal closure, independent of cuticular wax modulation.

In biological systems, adverse environmental conditions like drought stress intensify ROS production, posing risks to plant health. Antioxidant enzymes play a crucial role in mitigating ROS levels and reducing oxidative stress [29]. In Arabidopsis plants overexpressing *VvNAC17*, increased activities of SOD, POD, and CAT, alongside with elevated proline content and reduced MDA levels, were observed [30]. Similarly, our study with transgenic soybeans overexpressing *GmANKTM21* demonstrated enhanced drought tolerance, supported by elevated enzymatic activities of SOD, POD, and CAT, and decreased MDA levels (Figure 5).

To characterize the enriched pathways and genes affected by overexpressing *GmANKTM21*, we performed transcriptomic analyses in leaves and investigated significantly enriched biological processes and pathways by KEGG analysis. The results showed that differentially expressed genes were enriched into the MAPK signaling pathway (Figure 6A). The MAPK signaling cascade pathway is widely involved in multiple developmental stages of plants and has recently been shown to play a key role in abiotic stress signaling [31]. Recently, several MAPK family members have been reported to be directly or indirectly activated by the ABA pathway [32]. To validate the data, the levels of some differentially expressed genes were assessed by RT-qPCR in leaves exposed to drought stress (Figure 6B–G). The results showed that *GmANKTM21* regulated the expression of *GmSPK2.6*, *GmSPK4*, and *GmCYP707A1* under drought stress. Previous reports have indicated that *GmSPK2* and *GmSPK4* are induced under hypertonic and dehydration stress, while *GmCYP707A1* responds to both salt and dehydration stresses. Consequently, the RT-qPCR results confirmed that *GmANKTM21* regulates the expression levels of certain genes within the MAPK signaling pathway, thereby affecting plant drought tolerance.

## 4. Materials and Methods

### 4.1. Subcellular Localization 

The full-length cDNA sequence of GmANKTM21 was ligated to the GFP gene carrying the CaMV35S promoter, and the full-length cDNA sequence of GmANKTM21 was ligated to the GFP gene carrying the GmANKTM21 promoter. The recombinant vector was transferred into the Agrobacterium tumefaciens GV3101 strain, injected into tobacco epidermal cells, placed in a dark room at 25 °C for 12 h, and then restored to light conditions for 48 h of culturing and observed using laser confocal microscopy.

### 4.2. Plant Materials and Growth Conditions

Soybean (*Glycine max*) cultivars Dongnong50 and *GmANKTM21*-overexpressing transgenic soybean were used to analyze. Plants were grown in an artificial climate chamber under LD conditions (16-h:8-h light/dark) at 28 °C, relative humidity 55–65%, and light intensity 1000. On the 18th day after emergence, fully expanded trifoliate leaves were sampled 2 h after dawn from three individual plants for RT-qPCR analysis.

### 4.3. Vector Construction and Transformation

To generate the *GmANKTM21*-overexpressing transgenic soybean, the *GmANKTM21* coding DNA sequence (CDS) was amplified, as were the PCR fragments from Dongnong50 leaves, which contained *BamH* I and *Sac* I, respectively. The PCR product was then inserted downstream of the *CaMV 35S* promoter in the pBA-myc vector. The resulting construct was then introduced into *Agrobacterium* strain EHA105 and transformed into the Dongnong50 cultivar. The resulting construct also contained the selectable marker BAR for glufosinate resistance. T_0_, T_1_, and T_2_ plants were selected by smearing the expanded leaves with 160 mg·L^−1^ glufosinate. The glufosinate-resistant plants were used for RT-qPCR.

### 4.4. RNA Extraction and RT-qPCR Analysis

Total RNA was extracted from mature leaves of seedlings using the Ultrapure RNA Kit (CWBIO, Shanghai, China) according to the manufacturer’s instructions, with three independent biological replicates. Total RNA was reverse transcribed into first-strand cDNA using the Reverse Transcription System and SYBR Premix Top Mix (Transgen, Beijing, China). The measured Ct values were transformed into relative copy numbers utilizing the ∆∆Ct methodology. Relative fold enrichment was computed by normalizing the quantity of the target DNA fragment against the reference gene, *TUA5*. Three independent biological replicates (*n* = 3 plants) were collected and subjected to RT-qPCR with technical triplicates. Raw data underwent standardization as previously outlined. Details regarding the primers are provided in Appendix A.

### 4.5. Determination of Seed Germination Index

In this experiment, the relevant indexes of seed germination of *GmANKTM21*-OE and Dongnong50 soybean seeds under stress treatment were analyzed, and the seeds were treated with 0%, 7.5%, 15%, 22.5%, and 30% PEG6000 solution, put into the same volume of culture box, and grown for a total of seven days. Each group of experiments was subjected to three technical replicates and three biological replicates. The germination and growth indices are measured as follows:Germination rate = number of germinated seeds/number of tested seeds × 100%
Germination potential = number of germinated seeds within the specified date/number of seeds tested × 100%

The length of the hypocotyls of the plants was measured with a digital vernier caliper, and the electronic balance accurately weighed the fresh weight of the plant.

### 4.6. Leaf Water Loss Rate

*GmANKTM21*-OE and Dongnong50 leaves with similar shape and growth stage were selected to measure the fresh weight of the leaves at 0.5 h, 1 h, 2 h, 4 h and 5 h respectively and finally put into the oven to dry to a constant weight. Record the dry weight and calculate the leaf water loss rate. Six replications were performed for each group of experiments.

### 4.7. Stomatal Opening and Density

The stomatal opening and stomatal density of soybean with *GmANKTM21*-OE and Dongnong50, leaves with similar shape and growth stage were selected from normal irrigation and water interruption for one week, and the stomatal opening and stomatal density per unit area at different locations were measured under the microscope, and the average value was finally taken. The test was designed with three replications, and 100 stomata were examined in each replication.

### 4.8. Stratum Corneum Staining

*GmANKTM21*-OE and Dongnong50 leaves with similar shape and growth stage were selected. The materials were soaked in the same volume of 5% concentration toluidine blue solution freshly prepared, incubated at 37 °C for 2 h, and after the staining was completed, rinsed with distilled water many times and taken with a camera. Each group of experiments was subjected to three technical replicates and three biological replicates.

### 4.9. Wax Observation

*GmANKTM21*-OE and wild type (Dongnong50) leaves with similar shape and growth stage were selected and fixed in a fixative solution (75% ethanol, 5% acetic acid, 5% glycerol, 5% formaldehyde, and 10% deionized water) for at least one day, then the critical point was dried, sputtered gold-plated, and observed under SEM (JEM-6380LV, JEOL Ltd., Tokyo, Japan).

### 4.10. Determination of Antioxidant Enzyme Activity

The antioxidants, including superoxide dismutase (SOD), catalase (CAT), and peroxidase (POD), were extracted from approximately 0.1 g of the leaves of *GmANKTM21*-OE and Dongnong50 using a 1 mL extraction solution. The malondialdehyde (MDA) in the leaves was measured spectrophotometrically following the manufacturer’s instructions (Comin, Suzhou, China). Each recorded value represented three biological replicates and three technical replicates.

### 4.11. DAB and NBT Staining

The fully expanded leaves of *GmANKTM21*-OE and Dongnong50 after drought treatment for 6 days were immediately vacuum-infiltrated with diaminobenzidine tetrahydrochloride (DAB) solution (1 mg/mL 3,3′-diaminobenzidine-4HCl, pH 3.8) and NBT (nitroblue-tetrazole) solution (0.5 mg/mL NBT, 10 mM potassium phosphate, pH 7.8, 10 mM sodium azide), incubated in the dark for 8–10 h, and decolorized in 95% ethanol. Subsequently, the stained leaves were observed directly.

### 4.12. Expression Analysis of Downstream Genes by the GmANKTM21 Gene

The leaves and stem tips were taken at the R1 stage (the initial flowering stage of DN50), and biological replicates were performed three times for each sample. The samples were sequenced on the Illumina sequencing platform by Annoroad Biotechnology Co., Ltd. Differentially expressed genes were analyzed, and KEGG pathway analysis was performed. Subsequently, the differentially expressed genes were verified by RT-qPCR. All primers are listed in Appendix A.

## 5. Conclusions

In summary, we demonstrated the subcellular localization of GmANKTM21 in the cell membrane and nucleus. Overexpression of *GmANKTM21* in soybeans resulted in drought tolerance compared with wild-type (WT) plants, which was associated with stomatal closure and increased activity of antioxidant enzymes during drought stress. Additionally, KEGG and RT-qPCR analyses revealed that *GmANKTM21* positively regulates the expression of *GmSPK2* and *GmSPK4*, while negatively regulating the expression of *GmCYP707A1* in response to drought stress. These findings provide new insights into the function of *GmANKTM21* and may contribute to the enhancement of plant abiotic-stress tolerance through genetic manipulation in the future.

## Figures and Tables

**Figure 1 ijms-25-06972-f001:**
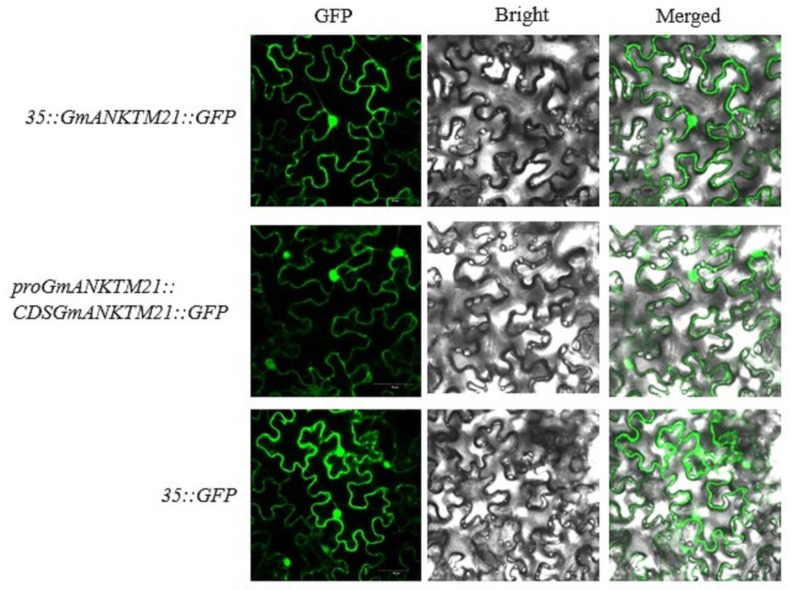
Subcellular localization of GmANKTM21 in tobacco leaf lower epidermal cells. Green fluorescent protein (GFP) fluorescence, fluorescence, bright field images, and merged images are displayed from left to right. Fluorescence was observed by confocal microscopy. Scale bar is 50 μm.

**Figure 2 ijms-25-06972-f002:**
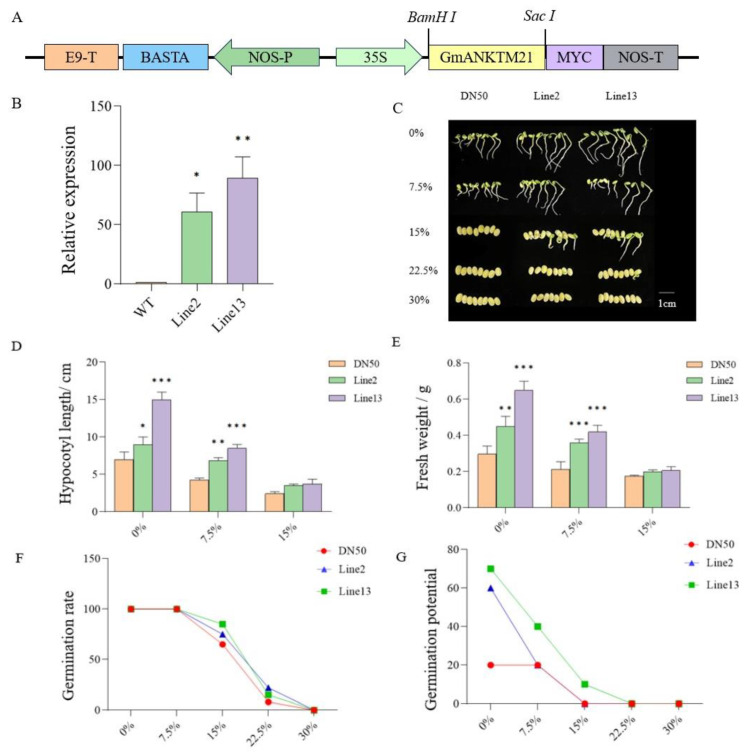
Germination of *GmANKTM21*-OE (Line2 and Line13) and wild-type (DN50) seeds under different concentrations of PEG6000. (**A**) T-DNA region map of recombinant vector; (**B**) RT-qPCR detection of *GmANKTM21* gene expression level in soybean overexpression plants. Values indicate the means ± SD (*n* = 3); (**C**) Germination phenotype of soybean seeds under drought; (**D**) Hypocotyl length; (**E**) Plant fresh weight, values indicate the means ± SD (*n* = 6); (**F**) Germination rate; (**G**) Germination potential. Values indicate the means ± SD (*n* = 6); the differences were statistically assessed using the Student’s *t* test; asterisks indicate significant differences (* *p* < 0.05, ** *p* < 0.01, *** *p* < 0.001) compared to those of the control samples (DN50).

**Figure 3 ijms-25-06972-f003:**
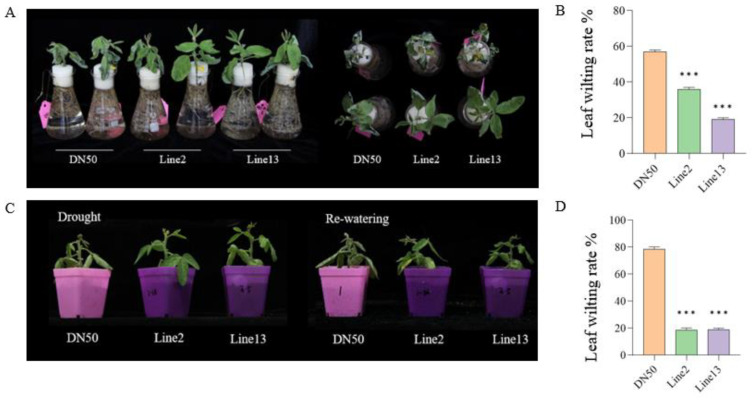
Seedling phenotype of soybeans with overexpression of the *GmANKTM21* gene under different drought simulation conditions. (**A**) PEG600 treatment; (**C**) Water cut-off treatment and restoration; (**B**,**D**) Leaf wilting rate. Values indicate the means ± SD (*n* = 6); asterisks indicate significant differences (*** *p* < 0.001) compared to that of the control samples (DN50).

**Figure 4 ijms-25-06972-f004:**
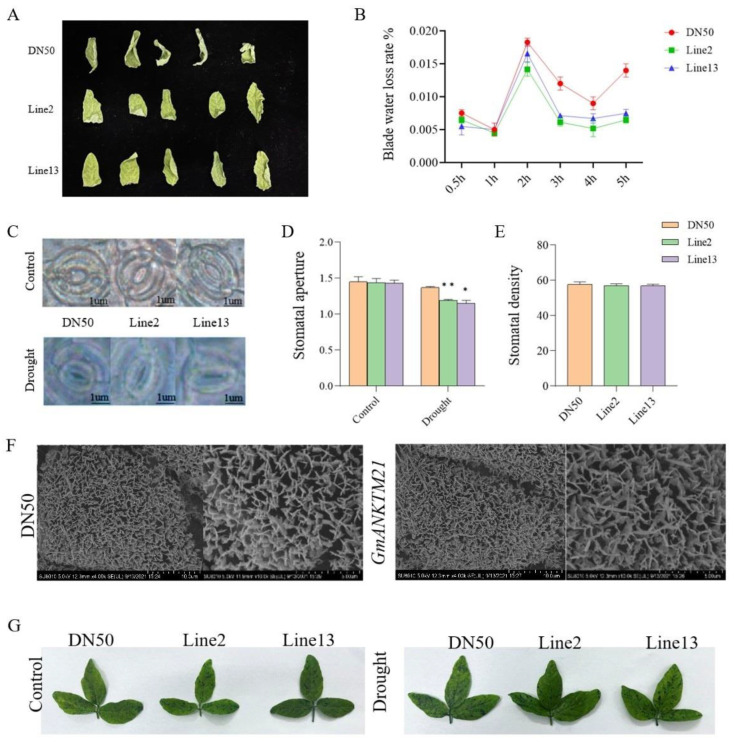
Water retention ability of plants overexpressing the *GmANKTM21* gene. (**A**) Leaf wilting phenotype; (**B**) Water loss rate; values indicate the means ± SD (*n* = 6); (**C**) Stomatal phenotype before and after drought treatment; (**D**) Stomatal opening. In total, 100 stomata were evaluated in each replicate; (**E**) Stomatal density; (**F**) Wax observation; (**G**) Toluidine blue staining. Values indicate the means ± SD (*n* = 6). The differences were statistically assessed using the Student’s *t* test; asterisks indicate significant differences (* *p* < 0.05, ** *p* < 0.01) compared to those of the control samples (DN50).

**Figure 5 ijms-25-06972-f005:**
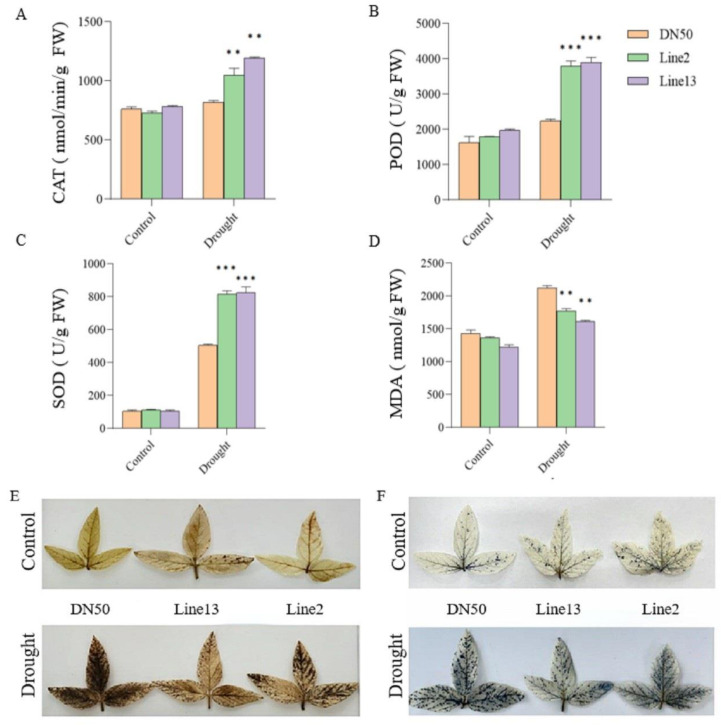
Analysis of physiological parameters of different soybean plants under drought treatment. (**A**) CAT activity; (**B**) POD activity; (**C**) SOD activity; (**D**) MDA content; (**E**) NBT staining; (**F**) DAB dyeing. Values indicate the means ± SD (*n* = 6). The differences were statistically assessed using the Student’s *t* test; asterisks indicate significant differences (** *p* < 0.01, *** *p* < 0.001) compared to those of the control samples (DN50).

**Figure 6 ijms-25-06972-f006:**
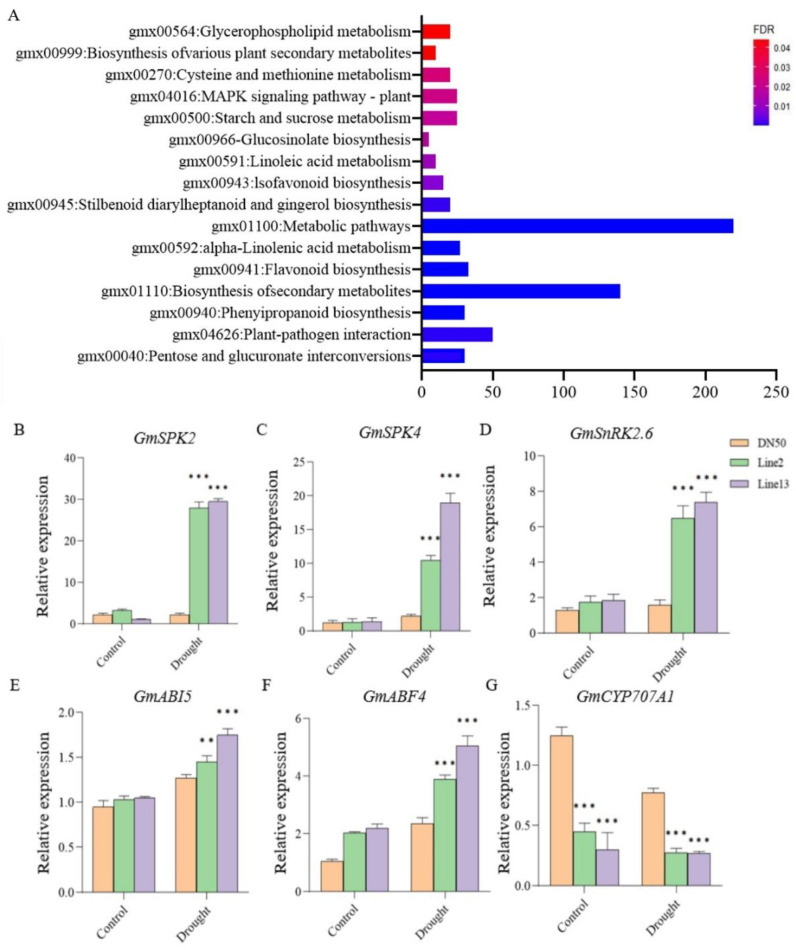
Analysis of the KEGG pathway and verification of drought tolerance genes. (**A**) KEGG pathway analysis of DEGs of *GmANKTM21*-OE and DN50 under the drought treatment; (**B**–**G**) Analysis of expression patterns of selected genes under the drought treatment by RT-qPCR. Values indicate the means ± SD (*n* = 3). The differences were statistically assessed using the Student’s *t* test; asterisks indicate significant differences (** *p* < 0.01, *** *p* < 0.001) compared to those of the control samples (DN50).

## Data Availability

The data presented in this study are available in the Appendix A.

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
