# Peer review of "GmANKTM21 Positively Regulates Drought Tolerance and Enhanced Stomatal Response through the MAPK Signaling Pathway in Soybean"

_ijms, 2024, doi:10.3390/ijms25136972_

Round 1
Reviewer 1 Report
Comments and Suggestions for Authors
The paper is interesting and the research well conducted. I only found a missing experiment. The overexpressed protein is presumably a membrane protein. According to figure 1b the level of overexpression is 50-100 fold higher. This could be a problem for membrane proteins as the transport to the proper membrane could saturate and the protein could be mislocalized or unproperly folded. In addition authors have shown that there is an important change in gene expression, determined by the RNA-seq experiments. Could it be that this change in expression would be due to the unfolded protein response? This may happen when you overexpress in such a high intensity membrane proteins. To clarify this point authors should tag the construction they are using with GFP or any other system and do some transient or stable expression experiments to find out the answer to what is happening:
a) Whether the protein is beign corretly processed.
b) In which membrane is located.
Minor points:
- A complete adress of the authors is missing.
- In each figure indicate which statistical analisys you have used.
- Line 229: mind the spacing after the stop.
Reviewer 2 Report
Comments and Suggestions for Authors
In this paper the authors presented some results how GmANKTM21 positively regulates drought tolerance.
They have investigated stomatal response through MAPK signaling pathway in soybean.
The study is interesting and novel.
However, the writing quality should be improved.
The discussion is too complex and wordy.
I have questions about the units of the enzymes and metabolites. Please use specific units. e.g. enzymes should be measured per mg protein basis.
Also, for POD it should not be metioned as U only.
MDA unit? fw or dw basis?
All should be carefully checked.
Discuss the ROS metabolism and antioxidant defense properly. It is superficially discussed. Cite more recent and relevant references.
Comments on the Quality of English LanguageCareful proofread is suggested.
Round 2
Reviewer 1 Report
Comments and Suggestions for Authors
The answer of the authors is quite inconclussive and I still see a major problem in the results. If the protein has a Transmembrane domain, how can it be located to the nucleus? If they did not suceed in doing a proper subcellular localization, at least include a western fractionating the soluble and the insolubel protein to check whether they have a full length protein properly located to the membrane, or free gfp resulting from protein degradation.
